# Multi-Task Learning and Improved TextRank for Knowledge Graph Completion

**DOI:** 10.3390/e24101495

**Published:** 2022-10-20

**Authors:** Hao Tian, Xiaoxiong Zhang, Yuhan Wang, Daojian Zeng

**Affiliations:** 1School of Computer Science, Nanjing University of Information Science and Technology, Nanjing 210044, China; 2The Sixty-Third Research Institute, National University of Defense Technology, Nanjing 210007, China; 3Science and Technology on Information Systems Engineering Laboratory, National University of Defense Technology, Changsha 410073, China; 4College of Information and Engineering, Hunan Normal University, Changsha 410000, China

**Keywords:** knowledge completion, a lite bidirectional encoder representations from transformers (ALBERT), multi-task learning, extractive summarization

## Abstract

Knowledge graph completion is an important technology for supplementing knowledge graphs and improving data quality. However, the existing knowledge graph completion methods ignore the features of triple relations, and the introduced entity description texts are long and redundant. To address these problems, this study proposes a multi-task learning and improved TextRank for knowledge graph completion (MIT-KGC) model. The key contexts are first extracted from redundant entity descriptions using the improved TextRank algorithm. Then, a lite bidirectional encoder representations from transformers (ALBERT) is used as the text encoder to reduce the parameters of the model. Subsequently, the multi-task learning method is utilized to fine-tune the model by effectively integrating the entity and relation features. Based on the datasets of WN18RR, FB15k-237, and DBpedia50k, experiments were conducted with the proposed model and the results showed that, compared with traditional methods, the mean rank (MR), top 10 hit ratio (Hit@10), and top three hit ratio (Hit@3) were enhanced by 38, 1.3%, and 1.9%, respectively, on WN18RR. Additionally, the MR and Hit@10 were increased by 23 and 0.7%, respectively, on FB15k-237. The model also improved the Hit@3 and the top one hit ratio (Hit@1) by 3.1% and 1.5% on the dataset DBpedia50k, respectively, verifying the validity of the model.

## 1. Introduction

In recent years, knowledge graphs (KGs) such as WordNet [1] and Freebase [2] have been widely used in many knowledge-intensive applications, including intelligent searching, question answering, dialogue systems, and recommender systems. Compared to traditional databases, KGs are more explicit and effective, and have a better searching ability. A KG schema is defined by ontologies that are often expressed as a group of concept definitions and hierarchical relationships between entity concepts. Ontology can restrict knowledge and ensure its quality. The storage form of a KG is a resource description framework (RDF), which is a knowledge representation framework based on a semantic web. RDF creates constraints on the values of nodes and edges, and formulates a unified standard. Based on the RDF, KGs can store a large amount of knowledge data in the triples that consist of a head entity, relation, and tail entity (h, r, t). Although KGs can contain billions of triples, most KGs, especially those constructed automatically, are still relatively incomplete owing to the rapid increase in real-world knowledge and tardy updating of KGs, which affects the quality of knowledge data and the efficiency of knowledge-intensive applications. To mitigate this problem, knowledge graph completion (KGC) technology has been studied in recent years. KGC is a part of knowledge processing in the construction of KGs, aiming to improve and enrich their structure and content.

The existing KGC technology can be divided into rule learning-based algorithms, path-based models, knowledge graph embedding models, and pre-trained language model-based approaches. In particular, pre-trained language model-based approaches use the pre-trained language model (PLM) to learn the text sequences of the triples and achieve a higher efficiency than other methods. Figure 1 shows an example of a KGC task.

However, these PLM-based approaches still have the following shortcomings: (1) difficulty in learning relations when dealing with lexically similar entities, (2) unnecessary and redundant information in the introduced entity descriptions, (3) time-consuming model training, caused by the large number of parameters of PLM.

To tackle the above problems, this study proposes a multi-task learning and improved TextRank for knowledge graph completion (MIT-KGC) model. The main components and modeling ideas are as follows: (1) To fully learn the relation information in KGs and more effectively predict the right entities from similar candidate entities, we combine the relation prediction, relevance ranking, and link prediction tasks into a multi-task learning framework based on MTL-DNN [3]. This framework can fuse relation and entity features, thereby overcoming the negative effects of lexically similar entities. (2) To avoid affecting downstream multi-task fine-tuning, we propose an improved extraction summary generation method based on TextRank [4]. By incorporating entity name coverage and sentence position information with the primary TextRank, we extract simplified description texts to alleviate the redundancy of entity descriptions. (3) A lite bidirectional encoder representations from transformers (ALBERT) [5], as the text encoder model, can decrease the number of parameters and improve the context learning ability compared to bidirectional encoder representations from transformers (BERT) [6]. ALBERT renders the word embedding dimension independently of the hidden dimension, through factorized embedding parameterization. In addition, the cross-layer parameter sharing of ALBERT prevents the parameter from increasing with network depth, making the computational complexity independent of the hidden size. ALBERT replaces the original next sequence prediction (NSP) task with a sequence order prediction (SOP) task, which trains the model to pay more attention to predicting sequence order instead of the text subject. 4) Furthermore, a feature enhancement component, including the mean-pooling strategy and bidirectional gated recurrent unit (BiGRU) [7], is utilized to reinforce the ability for excavating features. The mean-pooling strategy is a method for processing the embedding output and is adopted to improve the overall learning ability of ALBERT, because the original output strategy of ALBERT has an inadequate ability for sequence representation. BiGRU is a sequential neural network that is an improvement of the recurrent neural network (RNN) and can enhance the sequence feature information, owing to the parallel computation of ALBERT being weak in learning sequence positional information.

The contributions of this paper are summarized as follows:We propose a new KGC model named MIT-KGC that applies ALBERT, multi-task-learning, improved TextRank, mean-pooling strategy, and BiGRU. The model uses the improved TextRank to distill brief texts from entity descriptions and applies ALBERT to accelerate the training. The mean-pooling strategy and BiGRU are appended to enhance triple features, and multi-task learning is utilized to optimize the model for predicting the missing triples.We modify the traditional TextRank algorithm to make it more adaptive for KGC, by appending entity name coverage and sentence position information.Our method improves link prediction results, with MR, Hit@10, and Hit@3 increased by 38, 1.3%, and 1.9% on WN18RR [8], while MR and Hit@10 were enhanced by 23 and 0.7% on FB15K-237 [8]. Additionally, on the dataset DBpedia50k [9], Hit@3 and Hit@1 were increased by 3.1% and 1.5%, respectively, using our method.

The remainder of this paper is organized as follows: Section 2 introduces various methods for KGC, the TextRank-based algorithms, and the existing PLMs. In the last part of this section, we analyze the disadvantages of the existing research; Section 3 details the proposed MIT-KGC model, and describes the process of our model. Section 4 reports the experiment results and analysis. In addition, the dataset, baseline, experimental setting, experiment task, and evaluation metrics are also presented in this section. Section 5 provides the conclusions of our research and future work.

## 2. Related Work

### 2.1. Knowledge Graph Completion Model

The existing KGC models can be classified into four main categories: (1) Rule mining-based algorithms, such as AnyBurl [10] and DRUM [11], excavate rules from KGs and apply these rules for KGC tasks. However, rule searching and evaluation are usually time-consuming, which increases the ineffectiveness of these methods. (2) Path-based models, including path ranking approaches [12,13] and reinforcement learning-based models [14,15], tend to search paths linking head and tail entities. However, this still requires much time when searching multi-hop paths. (3) Knowledge graph embedding (KGE) models, such as TransE [16], TransH [17], DistMult [18], ComplEx [19] and RotatE [20], learn the embedding of entities and relations, to score the plausibility of triples for predicting the missing triples efficiently. In terms of efficiency regarding the above methods, rule mining-based algorithms and path-based models are more time-consuming than KGE models. Based on the survey in [19], TransE [16], DistMult [18], and ComplEx [19] have an approximate efficiency in time complexity, which is about one-sixth of TransH [17]. Nevertheless, these KGE methods cannot process complex relations very well and ignore external information. (4) Description-based models, such as DKRL [21], ConMask [22], and OWE [9], take advantage of entity descriptions and learn a transformation, to map the embeddings of an entity’s name and description to the graph-based embedding space. These models are generally proposed to address the open-world KGC task rather than the closed-world KGC task and are still based on KGE models. (5) Pre-trained language model-based approaches, such as KG-BERT [23] and MTL-BERT [24], apply PLMs for KGC, which use names or descriptions of entities and relations as input and fine-tune models to compute the plausibility scores of triples. PLM-based approaches achieve a higher efficiency and better performance than traditional methods. However, the redundancy of descriptions and neglect of multi-dimensional feature learning limits the precision of PLM-based approaches. More specifically, these models need an additional algorithm to extract briefer description texts, and require a multi-dimensional feature learning framework to combine entity and relation features.

### 2.2. Text Summarization Algorithm

TextRank [4], inspired by PageRank [25], is a classical graph-based sorting algorithm for realizing text summarization. Certain previous studies proposed improved algorithms based on TextRank. For example, Li et al. [26] made improvements to TextRank by adding text features. Researchers proposed studies [27,28,29,30] combining TextRank with different semantic analyses for keyword extraction. Bordoloi et al. [31] proposed a keyword extraction method based on supervised cumulative TextRank, emphasizing the correlation between words from three aspects: edge weight, damping coefficient, and interaction information. In addition, Liu et al. [32] used the subject model with the PageRank algorithm to extract keywords based on the importance of words. As the co-occurrence window can focus only on the correlation between local words, Zhou et al. [33] performed rough data reasoning on candidate keywords, thereby improving the accuracy of keyword extraction. Wang et al. [34] and Xiong et al. [35] extracted summaries of texts based on features such as inter-sentence similarity and sentence position. However, these methods do not adjust the TextRank algorithm for completing the KGs, and neglect the practical demands of the KGC task.

### 2.3. Pre-Trained Language Model

PLMs include BERT [6], GPT [36], and so on. These models are first pre-trained on large amounts of unlabeled text corpora with language modeling objectives, and then fine-tuned on specific downstream tasks, leading to a learning paradigm shift in natural language processing (NLP), making great contributions to NLP tasks. BERT is one of the classical PLMs, and some BERT-improved models such as RoBERTa [37] and ALBERT [5] have been proposed after BERT. Certain models such as KG-BERT [23] and MTL-BERT [24] have applied BERT for KGC tasks. However, the problem of long training time brought by BERT cannot be ignored. ALBERT [5] is a new lite BERT model for self-supervised learning of language representations. The core architecture of ALBERT is similar to BERT [6], but three specific innovations stand out: the factorization of embedding parameters, the sharing of parameters across layers, and the abandonment of the original NSP task and the use of SOP task. ALBERT achieves approximate precision in the same experiments and decreases runtime by using fewer parameters than BERT.

In summary, we conclude the following drawbacks of the related work: (1) the existing KGC models ignore the importance of learning entities and relations jointly, resulting in an impediment to recognizing correct entities; (2) the introduced description texts are mostly in large and redundant paragraphs, leading to inefficiency in using description texts and learning entities; (3) PLMs such as BERT and GPT are usually time-consuming when predicting triples, owing to their large number of parameters.

## 3. Proposed Method

To solve the problems of distinguishing similar entities, redundant entity descriptions, and slow model training, this study proposes a model called MIT-KGC. The structure of the model is illustrated in Figure 2.

The input of the model is divided into two parts: the description texts $D_{Head}$ and $D_{Tail}$ (Head/Tail Entity Description in Figure 2 and the relation’s name $R_{text}$. The MIT-KGC model consists of four parts: (1) text summarization: with the input texts composed of $D_{Head}$ and $D_{Tail}$, the TextRank-based extractive text summarization technology is utilized to extract the concise entity descriptions $H_{text}$ and $T_{text}$ from the original descriptions $D_{Head}$ and $D_{Tail}$; (2) sequence encoding: ALBERT first encodes the natural texts $D_{Head}$, $R_{text}$, and $D_{Tail}$, and special tags [CLS], [SEP] into initial embeddings ***H***, ***R***, ***T***, ***C***, and ***S.*** Subsequently, the initial embeddings are combined through certain rules as the input of ALBERT, and ALBERT is used to derive feature vectors to form the sematic feature matrix Z; (3) feature enhancement: the mean-pooling strategy is adopted to improve the encoded feature accuracy with the semantic feature matrix Z as the input. Furthermore, the output of the mean-pooling strategy $\tilde{E}$ is processed by BiGRU, to capture bidirectional semantic information; (4) multi-task learning: using the output of BiGRU E as the sharing hidden layer, a multi-task learning framework is designed to merge relation features into entity features using link prediction, relation prediction, and relevance ranking tasks, and the model is trained by optimizing the multi-task loss function (composed of Loss(LP), Loss(RP), and Loss(RR) in Figure 2) for KGC tasks.

### 3.1. Text Summarization

The purpose of the text summarization is to obtain concise descriptions from redundant and large paragraphs of entity descriptions. TextRank first preprocesses text word segmentation, and then identifies *n* text units (*n* is the number of sentences in an entity description) to create a graph model. Specifically, text units are used as the nodes of the graph, and sentence similarities are regarded as the edges of the graph. In this study, we follow the canonical TextRank to adopt the method based on word overlap as our similarity calculation method, as shown in Equation (1):(1)$$W_{s}(a,b)=\frac{\left|\{t_{k}|t_{k}\in Seq_{a}\wedge t_{k}\in Seq_{b}\}\right|}{\log(|Seq_{a}|)+\log(|Seq_{b}|)}$$
where $Seq_{a}$ and $Seq_{b}$ represent two sentences, |Seq| denotes the number of words of *Seq*.

After the similarity calculation, we can obtain the similarity feature matrix SD∈ℝn×n (a symmetric matrix consisting of *n* × *n* Ws). Sequentially, we initialize the weight value of each sentence equally to 1/*n* and obtain the sentence weight matrix B0=[1/n,1/n,…,1/n]. The weight values are iterated according to Equation (2), and finally we can obtain an iteration-completed sentence weight matrix Bf=[TR(Seq1),TR(Seq2),…,TR(Seqn)]:(2)TR(Seqi)=(1−d)+d∗∑Seqj∈In(Seqi)wji∑Seqk∈Out(Seqj)wjkTR(Seqj)
where * represents the multiplication, TR(Seqi) denotes the weight value of the *i*-th sentence, w∈SD is the weight of edges between nodes (sentence similarity), In(Seq) is the set of nodes pointing to node *Seq*, Out(Seq) is the set of nodes that *Seq* points to, and *d* is the damping ratio, representing the probability of one node jumping to another node (*d* = 0.85 in this paper).

Nevertheless, the canonical TextRank only takes the sentence similarity calculated by word overlap into consideration, and there are the following defects: (1) it neglects the importance of “entity name”, whereas the entity description we want often contains several “entity names”. Take the entity “Los Angeles” for instance, we want some descriptions such as “Los Angeles is the largest city in the western USA, and it is located in southern California”; (2) it ignores the effect of sentence position. In a redundant entity description, the front sentences are more likely to be the summative description texts. Therefore, we propose an improved TextRank algorithm to meet the needs of extracting simplified entity descriptions. We apply entity coverage and sentence position to adjust the final sentence weight values $B_{f}$, as shown in Equations (3) and (4):(3)$${W^{\prime }}_{e}(i)=\frac{\left|EN(Seq_{i})\right|}{\left|Seq_{i}\right|}$$
(4)$${W^{\prime }}_{p}(i)=1-\frac{i-1}{n}$$
where $EN(Seq_{i})$ denotes the number of entity names contained in *i*-th sentence, *n* is the number of sentences in the original description text, and *i* indicates the index of the sentence.

Through the calculation of entity coverage and sentence position, two corresponding feature matrices We′∈ℝn×1 and W′p∈ℝn×1 are obtained. We normalize them to obtain two normalized matrices, ***W_e_*** and ***W_p_*_,_** respectively. After that, according to Equation (5), the two normalized matrices are used to adjust Bf and make it more accurate and appropriate:(5)B=Bf∘(αWe+βWp)T
where ∘ represents the Hadamard product, α and β denote the weight of two normalized matrices, and α+β=1.

Finally, ranked by the scores of sentence weight values ***B***, the *x* (*x* = 1 in this paper) sentence with a higher score constitutes the entity description summarization. On the popular KGC datasets, most entities have their own descriptions (natural texts), and we apply the improved TextRank to purify these descriptions. For the very few entities with no descriptions, the improved TextRank is still used, but can only output one sentence (the name of the entity) for every entity.

### 3.2. Sequence Encoding

We extract the head and tail entity description summarization through the improved TextRank and then concatenate them with the relation text, to form the input sequence of ALBERT. Significantly, we append entity names to the head of the entity descriptions in the input sequence if it does not contain the entity names. The input sequence is separated by the special tags [CLS] and [SEP]. As an encoder, ALBERT aims to extract eigenvalues from triple texts and encode them into vector matrices with contextual semantic features.

ALBERT is a lightweight language model developed based on BERT. The number of albert-xlarge parameters used in this paper is 59 M, which is far smaller than the 108 M of bert-base, realizing a “thinner” model. Although it has fewer parameters, ALBERT can maintain a similar performance to BERT, benefiting from factorized embedding parameterization, cross-layer parameter sharing, and sentence-order prediction training tasks.

The main component of ALBERT is the transformer encoder, which is composed of several network layers stacked on each other. Each network layer is composed of a multi-head self-attention mechanism layer and a feed-forward network layer. In particular, the multi-head self-attention mechanism helps us capture the interrelation of words by calculating the attention matrices of several heads:(6)$$A(Q,K,V)=\mathrm{softmax}(\frac{QK^{T}}{\sqrt{d_{t}}})V$$
(7)$$head_{t}=A(QW_{t}^{Q},KW_{t}^{K},VW_{t}^{V}),t\in (1,2,3,\ldots ,h)$$
(8)$$MultiHead(Q,K,V)=\mathrm{Concat}(head_{1},head_{2},\ldots ,head_{h})W^{M}$$
where $W_{t}^{Q}$, $W_{t}^{K}$, $W_{t}^{V}$, $W^{M}$ are weight matrices, $d_{t}$ equals *H*/*h*, where *H* is the hidden size and *h* is the number of heads; and ***Q***, ***K***, and ***V*** are parameter matrices for query, key, and the value of self-attention mechanism, respectively.

By evaluating the relationship between all words, ALBERT adjusts the weight of each word in the sentence. After the text encoding by ALBERT, we obtain a new feature matrix Z∈ℝL×H (*L* is the length of input sequence and *H* is the hidden size of ALBERT) that integrates the deeper semantic features of the input context.

### 3.3. Feature Enhancement

Feature enhancement is composed of a mean-pooling strategy layer and BiGRU. A mean-pooling strategy layer is introduced to alleviate the problem of feature overlap and stacking. Considering the importance of [CLS] in previous surveys, we appropriately increase its proportion in the mean-pooling strategy instead of the absolute mean calculation. By contrast, BiGRU is used to facilitate the ability of the model to learn bidirectional sequence information. BiGRU uses the update gate to control the amount of information received from the previous time *t*-1 and applies the reset gate to decide how much information to ignore from the previous time *t*-1.

#### 3.3.1. Mean-Pooling Strategy

The feature matrix ***Z*** output by ALBERT is the input of the mean-pooling strategy layer. In this section, we first introduce the traditional [CLS] strategy and then explain the procedure of the mean-pooling strategy adopted in this study. An illustration of the traditional [CLS] strategy and the mean-pooling strategy is shown in Figure 3.

(1) The traditional [CLS] strategy assumes that the first hidden state in dimension *i* (i=1,2,3,…,H) of feature matrix ***Z*** is the [CLS] value h(i,1)∈ℝ1×1(i=1,2,3,…,H,j=1,2,3,…,L). Then, we merge the [CLS] value of each dimension to form a new feature matrix E′=(h(1,0),h(2,0),h(3,0),…,h(H,0)). Both BERT and ALBERT utilize the [CLS] output strategy to represent the input texts. Since [CLS] has no explicit semantic information, [CLS] can combine the semantic information of other words more fairly through multi-head self-attention mechanism layers. In Figure 3a, since the first line of the feature matrix is the hidden value of [CLS], we select all the values of the first line to obtain the representations of entities and relations. These representations can be effectively used to predict the missing parts of the triples from the viewpoint of link prediction.

However, the traditional [CLS] strategy may cause problems of feature overlap and stacking, which are more serious when the input texts are too long. Therefore, we propose a modified output strategy named the mean-pooling strategy.

(2) The mean-pooling strategy considers that [CLS] contains more important information than other words, and we do not simply apply a rigorous average value of each hidden dimension. In contrast, we introduce a hyper parameter μ∈[0,1] to increase the weight of the [CLS] information in the weighted hidden value hi¯∈ℝ1×1. Assuming that the hidden states in dimension *i* (i=1,2,3,…,H) of feature matrix ***Z*** are h(i,j)∈ℝ1×1(i=1,2,3,…,H,j=1,2,3,…,L), we calculate the weighted hidden value hi¯ using the different weights of the [CLS] information and other words. A portion of hi¯ is [CLS] information determined by the parameter μ, and the remaining is the mean of hidden values of other words. Afterwards, we merge the hi¯ of each dimension to form a new feature matrix, E˜∈ℝ1×H. The calculation of hi¯ and the new feature matrix E˜ are shown in Equations (9) and (10), respectively.
(9)hi¯=μh(i,1)+1−μL−1∑j=2Lh(i,j)
(10)E˜=(h¯1,h¯2,h¯3,…,h¯H)

#### 3.3.2. Bidirectional Gated Recurrent Unit

BiGRU takes the feature matrix $\tilde{E}$ (output of the mean-pooling strategy layer) as input, and the workflow at time *t* is as follows: (1) the reset gate coefficient $r_{t}\in [0,1]$ is calculated as Equation (11) by the input vector $e_{t}\in \tilde{E}$ at *t* step and the hidden state value $h_{t-1}$ of the previous GRU:(11)$$r_{t}=\sigma (h_{t-1}W_{r}+e_{t}W_{r}+b_{r})$$

The $h_{t-1}$ is selectively forgotten and updated to the candidate hidden state value ${\tilde{h}}_{t}$:(12)$${\tilde{h}}_{t}=\tanh((r_{t}⊙h_{t-1})W_{\tilde{h}}+e_{t}W_{\tilde{h}}+b_{\tilde{h}})$$

(2) Then, compute the update gate coefficient $z_{t}\in [0,1]$ as Equation (13) to select the important information of $e_{t}$ and $h_{t-1}$:(13)$$z_{t}=\sigma (h_{t-1}W_{z}+e_{t}W_{z}+b_{z})$$

Next, the update gate $z_{t}$ picks off the required information to update the hidden state value $h_{t}$:(14)$$h_{t}=(1-z_{t})⊙{\tilde{h}}_{t}+z_{t}⊙h_{t-1}$$

(3) After updating the hidden state value ht, the feature matrix E∈ℝ1×H, as the output of the feature enhancement layer, can be calculated as shown in Equation (15):(15)E=(h1,h2,…,ht,…,hH)

We set $W_{r}$, $W_{\tilde{h}}$, $W_{z}$ as weight matrices; $b_{r}$, $b_{\tilde{h}}$, $b_{z}$ are bias vectors; and ⊙ represents the multiplication of matrix elements.

### 3.4. Multi-Task Learning

The MIT-KGC model is optimized by a multi-task fine-tuning layer that contains three tasks (link prediction, relation prediction, and relevance ranking). We take the output E∈ℝ1×H of the feature enhancement layer as the sharing hidden layer of the multi-task fine-tuning layer. The link prediction task is regarded as the main target, and the relation prediction and relevance ranking are added to carry out multi-task learning and train the model’s ability to learn relation features and distinguish similar entities. During the training, a mini-batch is first selected from each epoch, and the three different loss functions are calculated for each task. Each loss function is then optimized according to the mini-batch stochastic gradient descent algorithm, to optimize the MIT-KGC model.

#### 3.4.1. Link Prediction

In this study, the link prediction task is regarded as a binary classification task, and the plausibility score of reasonable and correct triples should be higher. We assign each triple a positive score and select triple candidates with higher scores. Given an entity and a relation (h, r, ?) or (?, r, t), the link prediction task aims to predict another entity. The score function is set as SLP, as shown in Equation (16):(16)SLP=softmax(EWLP)
where WLP∈ℝH×2 is the parameter matrix of link prediction classification layer. SLP is a two-dimensional vector composed of two parts SLP1,SLP2∈[0,1] and SLP1+SLP2=1, representing the probability score of a triple belonging to two kinds of label.

Since the triples in the dataset are all facts, which constitute the positive sample set $D^{+}$, a substitution method is needed to construct a negative sample set $D^{-}$, as shown in Equation (17):(17)$$D^{-}=\{(h^{\prime },r,t)|h^{\prime }\in E\wedge h^{\prime }\neq h\wedge (h^{\prime },r,t)\notin D^{+}\}\cup \{(h,r,t^{\prime })|t^{\prime }\in E\wedge t^{\prime }\neq t\wedge (h,r,t^{\prime })\notin D^{+}\}$$

Thus, given positive and negative sample sets $D^{+}$ and $D^{-}$, the binary cross entropy loss function $L_{LP}$ of link prediction is shown in Equation (18):
(18)$$L_{LP}\;=\;-\sum_{T\in D^{+}\cup D^{-}}\left((1-y_{T})\log(S_{LP2})+y_{T}\log(S_{LP1})\right)$$
where $y_{T}\in \{0,1\}$ is the label of triple (negative or positive sample).

#### 3.4.2. Relation Prediction

The purpose of relation prediction is to infer the missing relations using two given entities (h, ?, t). Relation prediction trains the model to predict the covered relations from known entities to learn relation features. The essence of relation prediction is a binary classification task similar to link prediction, and which aims to increase the score of correct relations. The score function of the relation prediction SRP and the cross entropy loss function LRP are shown in Equations (19) and (20):(19)SRP=softmax(EWRP)
(20)LRP = −∑T∈D+yRlog(SRP)
where WRP∈ℝH×R is the relation prediction classification layer parameter matrix, *R* is the number of relations in the dataset, and yR is the relation label.

#### 3.4.3. Relevance Ranking

The purpose of relevance ranking is to give higher scores to the correct entities and train the model to differentiate reasonable entities from non-reasonable entities, to overcome the influence brought by similar entities. The score function of relevance ranking SRR is shown in Equation (21):(21)SRR=sigmoid(EWRR)
where WRR∈ℝH×1 is the parameter matrix of the relevance ranking task.

Differently from the above two tasks, margin ranking loss is used to optimize the distance of different entities, as shown in Equation (22):
(22)$$L_{RR}\;=\;\sum_{T\in D^{+},T^{\prime }\in D^{-}}\max\{0,S_{RR}^{N}-S_{RR}^{P}+\lambda \}$$
where $S_{RR}^{N}$ denotes the negative sample score function calculated using Equation (21), $S_{RR}^{P}$ is the positive one, and λ is the margin of the margin ranking loss function.

## 4. Experiment and Analysis

### 4.1. Dataset

The datasets used in this paper were FB15K-237 [8], WN18RR [8], and DBpedia50k [9], which are the most popular KGC datasets. WN18RR is a subset of WordNet [1], containing English triples and entity description information. FB15k-237 is a subset of FreeBase [2] and contains more complex English entity relations and description texts than WN18RR does. DBpedia50k is a dataset for both open-world and closed-world KGC tasks. It provides long and detailed descriptions of the entities. Table 1 lists the statistics of the datasets used in this study.

### 4.2. Baseline

The baseline models in this paper were divided into PLM-based KGC models, traditional KGC models, and description-based KGC models. The PLM-based KGC models include LP-BERT [38], MTL-BERT [24], KG-XLnet [39], and KG-BERT [23]. Traditional KGC models include RESCAL-N3-RP [40], DensE [41], R-GCN [42], RotatE [20], ConvE [8], ComplEx [19], DistMult [18], and TransE [16]. In addition, we also used three description-based KGC models DKRL [21], ConMask [22], and OWE [9] as the traditional baseline models.

### 4.3. Experimental Setting

Albert-xlarge was selected as the text encoder of MIT-KGC in this study. We used the Adam optimizer for training and tuning the hyper parameters of our model. The maximum sentence length was 128 for all three datasets. We set the minibatch size = 64 for FB15k-237, 32 for WN18RR and DBpedia50k; the training epoch = 7 for FB15k-237 and WN18RR, and 6 for DBpedia50k; learning rate = 5 × 10^−5^, and margin of the loss function = 0.1. In addition, we set the [CLS] weight parameter μ=0.2 in the mean-pooling strategy.

### 4.4. Experiment Task and Evaluation Metrics

This paper studies a typical task of knowledge completion, link prediction, the goal of which is to predict the missing entity according to another entity and the relation between them. The main evaluation metrics were the mean rank (MR) and top-*k* hit ratio (Hit@*k*). MR refers to the average ranking of the target triples. The smaller the MR is, the better the model performance is. Hit@*k* calculates the proportion of the correct triples ranked among the top *k*, and a larger Hit@*k* indicates a better model performance. The experiment excluded the influence of other correct triples after replacement [16].

### 4.5. Link Prediction Result

The link prediction results of the competing models and our model on datasets FB15K-237 and WN18RR are shown in Table 2. The result figures were taken from the original papers, and the bold numbers denote the state-of-the-art performance of each matrix. The experimental results showed that the MIT-KGC model surpassed all other approaches on MR, Hit@10, and Hit@3.

On the FB15K-237 dataset, the proposed model made some experimental progress, with MR and Hit@3 increased by 23 and 0.7%, respectively. The improvement was especially significant in terms of MR. The reason for this may be that FB15K-237 has many complex relations. Multi-task learning can be used to effectively learn these relations. Moreover, entity description texts are sufficiently long for text summarization technology to reduce redundant texts and make it easier to predict correct entities.

Furthermore, on the WN18RR dataset, MIT-KGC outperformed the other models, by increasing MR, Hit@10, and Hit@3 by 38, 1.3%, and 1.9%, respectively, among which MR increased most significantly. Compared with the FB15k-237 dataset, the improvement brought by MIT-KGC on WN18RR was more remarkable. This reason for this may be that there are more lexically similar entities on WN18RR than FB15k-237, so that multi-task learning can facilitate the ability to pick out similar candidates and elevate the score of correct candidates. Additionally, the mean of the node degree distribution of dataset WN18RR is smaller than FB15k-237 [43], in other words, dataset WN18RR is sparser in graph structure than FB15k-237. Due to PLMs’ outstanding ability in understanding semantics, the ALBERT we applied in MIT-KGC was more competent in processing the sparser data of WN18RR. Consequently, the experimental results on dataset WN18RR were more noticeably improved than those on the FB15k-237.

However, we found that the Hit@1 result was significantly worse than the translation distance models DensE [41] and RESCAL-N3-RP [40] on datasets WN18RR and FB15k-237. At the same time, regarding the metric Hit@3 on FB15k-237, MIT-KGC also ranked second and was slightly behind the SOTA model RESCAL-N3-RP [40], by 0.8%. This is because the PLM is mainly modelled from the semantic level and lacks the structural features of triples. Consequently, it seems more difficult for MIT-KGC to predict the correct entity in the first place. Although some translation distance models perform better than MIT-KGC on the Hit@1 result, they cannot understand the text semantics. If some entities are not ever seen during the prediction, the inductive performance of translation distance models will be poor. On the contrary, the PLMs are more reliable, which is why MIT-KGC can far exceed the translation distance models, to achieve state-of-art performance on MR and Hit@10.

In addition, we compared our model with several description-based models. As shown in Table 3, our model surpassed the three description-based models on Hit@3 and Hit@1, and ranked second on MR and Hit@10. Differently from DKRL [21], ConMask [22], and OWE [9], our model applies PLM rather than traditional KGE models to encode the triples and obtain the text embeddings. That is why MIT-KGC improved the Hit@3 and Hit@1 by 1.7% and 0.8%. Moreover, the traditional description-based models are designed to learn novel triples by relying on external text-enhanced information, but our model is more robust to these unseen triples, because of the strong semantic learning ability of PLM.

Although our model performed well on Hit@3 and Hit@1, it still fell behind in the other two metrics. ConMask [22] achieved the best results on MR and Hit@10, by benefiting from its relationship-dependent content masking and target fusion mechanism. We suppose that MIT-KGC is inferior to ConMask on MR and Hit@10 because our model does not effectively locate the key information related to the prediction task from the introduced texts. However, considering the comprehensive performance on all three datasets, MIT-KGC generally made clear progress.

### 4.6. Ablation Experiments

#### 4.6.1. Training Tasks Strategy Experiment

To analyze the different effects of training tasks in multi-task learning framework, we design different combinations of link prediction (LP), relation prediction (RP), and relevance ranking (RR). The experimental results of different training task strategies are shown in Table 4.

According to the results, the “LP + RP + RR” task strategy used in this paper achieved the best result. On dataset WN18RR, compared with only being trained by LP, “LP + RP + RR” improves matrices by 31.1, 12.1%, 11.1%, and 9.2%, showing that the multi-task learning is beneficial to the overall performance. From the analysis of the experimental results of the “LP + RP” and “LP + RR” task strategies, the former improved MR, Hit@10, Hit@3, and Hit@1 by 23.2, 8.7%, 7.3%, and 3.9% and the latter by 2.8, 1.8%, 3.0%, and 2.8%, indicating that the added RP and RR were effective and valid. Moreover, we found that RR resulted in an obvious improvement over RP. This is because RR helped our model distinguish corrupted entities from the correct ones, leading to higher scores and ranks for target entities.

#### 4.6.2. Encoder Model Analysis

To compare the experimental performances and training runtime of the different encoders, we designed another KGC model with BERT as an encoder, specifically bert-xlarge (from ALBERT) and bert-large (from BERT). This paper compared BERT-based (bert-xlarge and bert-large) with ALBERT-based (albert-xlarge and albert-large) models, and the main parameters of the four encoders are shown in Table 5. We did not consider albert-xxlarge as an encoder, because of its too high computational complexity, although it may perform well. The experimental results and running speed on dataset WN18RR are shown in Table 6, and we regard bert-xlarge as 1.0× speed, owing to it having the longest runtime.

This shows that albert-xlarge improved by 10.3, 6.9%, 5.7%, and 2.2% on MR, Hit@10, Hit@3 and Hit@1, while reaching a speed 2.1-times faster than bert-xlarge. This is because albert-xlarge reduces the parameters and increases the data throughput with the aid of factorized embedding parameterization and cross-layer parameter sharing. Meanwhile, albert-xlarge keeps the embedding size unchanged through factorized embedding parameterization, to strengthen the ability for model understanding by enlarging the hidden size. From the perspective of speed, albert-large ranked first but performed worse than bert-large in our experiment. Considering the time cost and prediction accuracy in a balanced way, we validly applied albert-xlarge as our encoder because of its medium training speed and excellent prediction performance.

#### 4.6.3. Text Summarization Analysis

We analyzed the improved TextRank using three aspects: link prediction results, case studies, and text length changes. As shown in Table 7, MIT-KGC improved MR, Hit@10, Hit@3, and Hit@1 by 14.9%, 5.8%, 5.8%, and 5.7%, respectively, compared with the model without TextRank. We also compared the improved TextRank with the original TextRank algorithm. MIT-KGC using the improved TextRank could increase MR, Hit@10, Hit@3, and Hit@1 by 9.2%, 3.4%, 3.7%, and 4.4% compared to the original TextRank. This result indicates that the improved TextRank was positively related to the link prediction accuracy, and our modifications to the original TextRank were valid.

To more specifically reveal the achievement of the improved TextRank, as shown in Table 8, we tracked four entities from datasets (the first two are from WN18RR, and the latter two are from FB15k-237), to observe their changes in the text description. Meanwhile, we selected four corresponding triples that contained the four entities, to observe their prediction results. When extracting the descriptions, in addition to the sentence similarity, we also considered the influence of entity coverage and sentence position. As an example of “protective”, the first sentence of the paragraph, “protective, intended or adapted to afford protection of some kind;” has the same entity coverage as other sentences but ranks first in terms of sentence position. Synthetically calculating the entity coverage, sentence position, and sentence similarity, we could obtain a weight-based rank of sentences and take the sentence that ranks first as the extracted description. Similarly, other entities can acquire a brief extracted description.

Moreover, we investigated the prediction results of test triples and found that the results were consistent with our expectations (italic in Table 8). For instance, the model predicted the right head entity “United Kingdom” for the incomplete triple (?, /location/location/contains, Halifax), and the similar but invalid entities “United States of America” and “London” ranked behind. Therefore, our model has a good ability to pick out the correct entities from other similar entities.

In addition, we studied the length change of the entity description on the FB15k-237 and WN18RR datasets from the results shown in Table 9. After being processed by improved TextRank, the average length of entity description on FB15k-237 was decreased by 692.3 (80.1%), while that on WN18RR was decreased by 25.1 (28.0%). The text summarization algorithm greatly reduced the redundancy and improved the quality of description texts. Moreover, since there are more complex and long text descriptions in FB15k-237, the length change of the description decreased more obviously on FB15k-237 than on WN18RR.

#### 4.6.4. Feature Enhancement Component Experiment

Besides the above experiments, we also conducted an ablation experiment for feature enhancement components (BiGRU and mean-pooling strategy) of MIT-KGC, as shown in Table 10.

The results in Table 10 demonstrate that our model, MIT-KGC, performed better than the ablated models removed by BiGRU or mean-pooling strategy on WN18RR. Specifically, with BiGRU, MIT-KGC improved on the experimental results, with increases of 25.8, 6.9%, 9.1%, and 4.8% on MR, Hit@10, Hit@3, and Hit@1. If the mean-pooling strategy was removed, the model would decrease the metrics by 37.3, 10.7%, 14.2%, and 11.6%. The improvement illustrates that the introduced BiGRU and mean-pooling strategy both contributed to enhancing features. In particular, compared with BiGRU, the mean-pooling strategy contributed more to facilitating the link prediction performance by improving the encoding ability of ALBERT. In general, each component plays a pivotal role in MIT-KGC.

## 5. Conclusions

In this study, addressing the problems of the existing KGC models, such as the lack of relations and a similar entity learning ability, the difficulty in processing redundant entity description texts, and the problem of long training times, we proposed an effective MIT-KGC model for link prediction. Specifically, we propose an improved TextRank to address the redundant information in entity descriptions. Meanwhile, we considered ALBERT as our encoder model, because it has fewer parameters and a higher operation efficiency than BERT. Moreover, we introduced a mean-pooling strategy to enhance the expression ability of ALBERT and applied BiGRU to study the sequence information, while the ability to learn relations was improved using the multi-task learning framework.

The experimental results showed that compared with the baseline models, MIT-KGC achieved the best performance on MR, Hit@10, and Hit@3 on WN18RR; MR and Hit@10 on FB15K-237; and Hit@3 and Hit@1 on DBpedia50k, demonstrating the effectiveness of MIT-KGC. Moreover, our ablation experiments indicated that the “LP + RP + RR” task strategy is valid and positive for the performance of the MIT-KGC model. In addition, the encoder model experiment analysis showed that ALBERT accelerated the speed of our model and improved the prediction results. Meanwhile, the results of the text summarization analysis indicated the positive influence of the improved TextRank on link prediction and refined descriptions, and the case study demonstrated the model’s ability to distinguish similar entities. Furthermore, we conducted a feature enhancement components experiment to demonstrate the significance of the mean-pooling strategy and BiGRU.

However, some flaws still exist in our model. MIT-KGC did not perform the best on Hit@1. In future studies, we will explore how to improve the results on Hit@1 and adopt more advanced text summarization technology to extract or generate the required texts from entity descriptions.

## Figures and Tables

**Figure 1 entropy-24-01495-f001:**
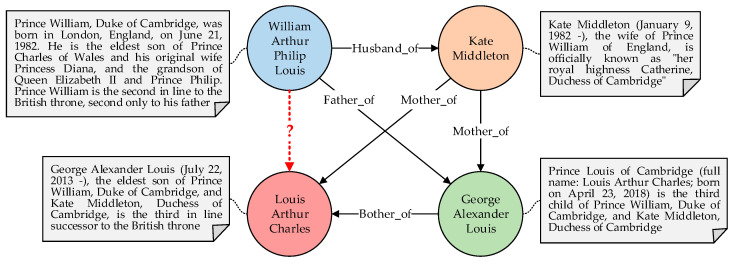
An example of a KGC task. The circles represent entities, the lines between circles denote relations, and the texts linked with circles refer to entity descriptions. Given the entities and relations, KGC is utilized to learn semantic information of existing triples and precisely infer the missing relation represented by the red dotted line.

**Figure 2 entropy-24-01495-f002:**
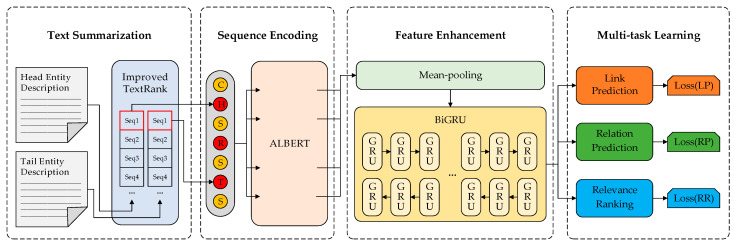
Overall structure of the MIT-KGC model. C, H, S, R, and T represent the initial embeddings of [CLS], head entities, relations, [SEP], and tail entities, respectively. Seq1 denotes the description sentence that ranks first after text summarization.

**Figure 3 entropy-24-01495-f003:**
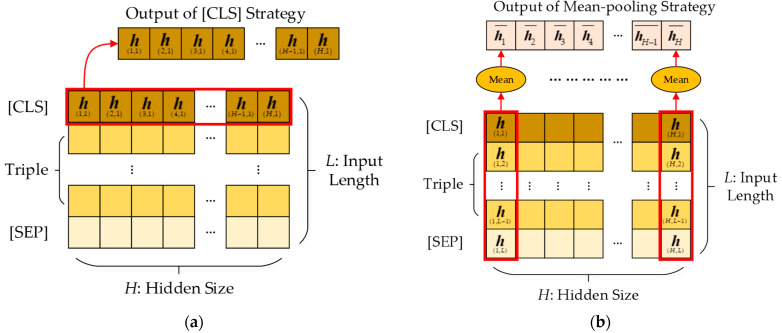
The computation procedure of the traditional [CLS] strategy and the mean-pooling strategy. The squares in the two pictures are hidden states value (dark, medium, and light brown, respectively, represent hidden values of the [CLS] tag, triple, and [SEP] tag). The ovals with “Mean” refer to the weighted calculation of hidden values (as shown in Equation (9)). (**a**) The traditional [CLS] strategy simply takes the first line of the primary feature matrix, the hidden value of [CLS] tag as the output feature matrix. (**b**) The mean-pooling strategy first calculates the weighted value of every hidden dimension, and then constructs the output feature matrix by combining the weighted values.

**Table 1 entropy-24-01495-t001:** Statistics of the datasets.

Dataset	Entities	Relations	Train	Validation	Test
FB15k-237	14,541	237	272,115	17,535	20,466
WN18RR	40,943	11	86,835	3034	3134
DBpedia50k	24,624	351	32,388	123	2095

**Table 2 entropy-24-01495-t002:** Link prediction results of the models. The first three compared baselines are PLM-based KGC models, and the last are traditional KGC models. The bold numbers refer to the best results in that metric.

Model	FB15k-237				WN18RR			
	MR	Hit@10(%)	Hit@3(%)	Hit@1(%)	MR	Hit@10(%)	Hit@3(%)	Hit@1(%)
MIT-KGC(ours)	**109**	**57.5**	41.7	21.2	**51**	**76.5**	**58.2**	33.5
LP-BERT(2022)	154	49.0	33.6	22.3	92	75.2	56.3	34.3
MTL-BERT(2020)	132	45.8	29.8	17.2	89	59.7	38.3	20.3
KG-XLNet(2021)	-	-	-	-	108	51.8	-	-
KG-BERT(2019)	153	42.0	-	-	97	52.4	30.2	4.1
RESCAL-N3-RP(2021)	163	56.8	**42.5**	**29.8**	-	58.0	50.5	**44.3**
DensE(2020)	169	53.5	38.4	25.6	3052	57.9	50.8	**44.3**
R-GCN(2018)	600	30.0	18.1	10.0	6700	20.7	13.7	8.0
RotatE(2018)	177	53.3	37.5	24.1	3340	57.1	49.2	42.8
ConvE(2018)	245	49.7	34.1	22.5	4464	53.1	47	41.9
ComplEx(2016)	546	45.0	29.7	19.4	7882	53	46.9	40.9
DistMult(2014)	512	44.6	30.1	19.9	5110	49	44	39
TransE(2013)	323	44.1	37.6	19.8	3384	50.1	-	-

**Table 3 entropy-24-01495-t003:** Link prediction results of the models. The three compared baselines are description-based KGC models. The bold numbers refer to the best results in that metric.

Model	DBpedia50k			
	MR	Hit@10(%)	Hit@3(%)	Hit@1(%)
MIT-KGC(ours)	43	79.8	**68.3**	**53.4**
OWE(2019)	-	76.0	65.2	51.9
ConMask(2018)	**16**	**81.0**	64.5	47.1
DKRL(2016)	70	40.0	-	-

**Table 4 entropy-24-01495-t004:** Link prediction results of different training tasks strategies. The bold numbers refer to the best results in that metric.

Training Tasks	WN18RR			
	MR	Hit@10(%)	Hit@3(%)	Hit@1(%)
LP + RP + RR	**51.4**	**76.5**	**58.2**	**33.5**
LP + RP	74.6	67.8	50.9	29.6
LP + RR	54.2	74.7	55.2	30.7
LP	82.5	64.4	47.1	24.3

**Table 5 entropy-24-01495-t005:** Parameters of different encoders.

Model	Type	Layers	Hidden	Embedding
ALBERT	large	24	1024	128
	xlarge	24	2048	128
BERT	large	24	1024	1024
	xlarge	24	2048	2048

**Table 6 entropy-24-01495-t006:** Experimental results of the different encoders. The bold numbers refer to the best results in that metric, and the fastest training speeds are underlined.

Encoder	WN18RR				
	MR	Hit@10(%)	Hit@3(%)	Hit@1(%)	Speed
${\mathrm{ALBERT}}_{xlarge}$	**51.4**	**76.5**	**58.2**	**33.5**	2.1×
${\mathrm{ALBERT}}_{large}$	92.4	65.9	43.1	23.0	6.2×
${\mathrm{BERT}}_{xlarge}$	175.5	49.7	22.4	11.7	1.0×
${\mathrm{BERT}}_{large}$	61.7	69.6	52.5	31.3	3.4×

**Table 7 entropy-24-01495-t007:** Ablation experiment with the improved TextRank. The bold numbers refer to the best results in that metric.

Models	WN18RR			
	MR	Hit@10(%)	Hit@3(%)	Hit@1(%)
MIT-KGC (improved TextRank)	**51.4**	**76.5**	**58.2**	**33.5**
MIT-KGC (original TextRank)	60.6	73.1	54.5	29.1
MIT-KGC (without TextRank)	66.3	70.7	52.4	27.8

**Table 8 entropy-24-01495-t008:** Case study of entity description. We divide the entities and relations using square brackets, and the missing entities are in italics in the fourth column. The predicted entities with the highest scores are listed in the fifth column and the correct entities are marked in italics.

Entities	Description	Extracted Description	Test Triples	Predicted Entities
protective 01887076	protective, intended or adapted to afford protection of some kind; “a protective covering”; “the use of protective masks and equipment”; “protective coatings”; “kept the drunken sailor in protective custody”; “animals with protective coloring”; “protective tariffs”	protective, intended or adapted to afford protection of some kind.	[*preventive*] [_also_see] [protective]	[*preventive*] [unarmoured] [protectiveness] ……
element 03081021	element, an artifact that is one of the individual parts of which a composite entity is made up; especially a part that can be separated from or attached to a system; “spare components for cars”; “a component or constituent element of a system”	element, an artifact that is one of the individual parts of which a composite entity is made up.	[*supplement*] [_hypernym] [element]	[*supplement*] [crystal] [oxide] ……
Halifax /m/0cdw6	Halifax is a Minster town, within the Metropolitan Borough of Calderdale in West Yorkshire, England. It has an urban area population of 82,056 in the 2001 Census. It is well known as a centre of England’s woollen manufacture from the 15th century onward, originally dealing through the Halifax Piece Hall. Halifax is known for its Mackintosh chocolate and toffee, the Halifax bank, and the nearby Shibden Hall.	Halifax is a Minster town, within the Metropolitan Borough of Calderdale in West Yorkshire, England.	[*United Kingdom*] [/location/location/contains] [Halifax]	[*United Kingdom*] [United States of America] [London] ……
Sandra Bernhard /m/0m68w	Sandra Bernhard is an American comedian, singer, actress and author. She first gained attention in the late 1970s with her stand-up comedy in which she often bitterly critiques celebrity culture and political figures. Bernhard is number 97 on Comedy Central’s list of the 100 greatest standups of all time.	Sandra Bernhard is an American comedian, singer, actress and author.	[Sandra Bernhard] [/people/person/profession] [*Actor-GB*]	[*Actor-GB*] [Professor-GB] [Film Director] ……

**Table 9 entropy-24-01495-t009:** Length change of the entity descriptions. The lengths are measured by character number.

Dataset	Summarization	Shortest Description	Longest Description	Average Length
FB15k-237	No	25	4019	864.8
	Yes	8	1019	172.5
WN18RR	No	9	534	89.8
	Yes	9	519	64.7

**Table 10 entropy-24-01495-t010:** Experimental results of the feature enhancement components. The bold numbers refer to the best results in that metric.

Models	WN18RR			
	MR	Hit@10(%)	Hit@3(%)	Hit@1(%)
MIT-KGC	**51.4**	**76.5**	**58.2**	**33.5**
-BiGRU	77.2	69.6	49.1	28.7
-Mean-pooling	88.7	65.8	44.0	21.9

## Data Availability

The datasets WN18RR and FB15K-237 investigated in this work are publicly available at https://github.com/yao8839836/kg-bert/tree/master/data (accessed on 4 September 2019), and the public dataset DBpedia50k can be found at https://github.com/haseebs/OWE (accessed on 13 January 2021).

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
