# Peer review of "Multi-Task Learning and Improved TextRank for Knowledge Graph Completion"

_entropy, 2022, doi:10.3390/e24101495_

Round 1

Reviewer 1 Report

The paper presents a novel architecture for knowledge graph construction using special integration of known tools in knowledge engineering, NLP.  The research topic is relevant, it focuses on an important problem domain. The structure of the paper is appropriate; the model involves current technologies. The paper contains an experimental test section to compare the different parameter settings of the proposed algorithm. It has no direct test comparison with other methods; this fact  limits the extensibility of the results. The paper presents a good description of the domain.

On the other hand, the current version has some shortcomings (especially regarding formal description, explanation of the theoretical foundation), thus I suggest some revision of the paper. I can summarize my comments in the following points.

  1. Extend the formal description of the proposed architecture with a higher level description which includes the input/output data items of the main modules (see Figure 2).  This level is more important for the readers than the selected formulas  on the internal functions as the main novelty here is the specific integration of existing units.

  2. I suggest adding some explanation about the benefits of [CLS] strategy, why is it useful to return the first line corresponding to the CLS tag; why is this value relevant from the viewpoint of linking prediction?    

  3. Regarding equation 16, how can we use a layer with 2 outputs to predict the related entity (number of entities is very large)?

  4. Is it possible to show the positive effect of proposed modification in textrank (equation 5)?

  5. The output of textrank is a reduced list of sentences. How can it be used to extract head and tail entities? 

  6. The standard train and test data files used in the presented tests (like FB15k, WN18RR, DPpedia) contain triplets and not natural texts. It is not explained directly in which cases are the NLP units (like text summarization)  used and when not.     

  7. The meaning of some formal symbols are not explained, like W’ and W on page  6; W^Q,,W^K,V,... on page 6, S_RR,S’_RR on page 9

  8. The usage of some symbols is not unambiguous, like   a) a sentence is given with symbol seq or X (Tr(X_i)) (page 5); is h equal to k in (page 5, eq. 7)? 

  9. Add  a brief description in the introduction on the ontology, RDF background of the triplets

  10. Try to avoid splitting a table to pages.

  11. I suggest to extend the literature survey with some lines to give a general comparison of the efficiency of the  key alternative methods (like TransE)

Reviewer 2 Report

This paper deals with Knowledge graph completion, an important technology to improve data quality, by proposing MIT-KGC (Multi-task-learning and Improved TextRank for Knowledge Graph Completion), a multi-task-learning and Improved TextRank for Knowledge Graph Completion model. The WN18RR and FB15k-237 have been used for the experiments of the proposed model. The improved TextRank summarization technique

The paper is well-structured and lucidly written. A detailed account of the process and the techniques involved is provided. The results, including the ablation study, are reported as objectively as possible. The confirmatory dataset is adequately identified and reported. The conclusions are sufficiently backed up by evidence. For all these reasons, I recommend this paper for publication, after the following minor corrections have been made:

Minor corrections:

Line 34 has should be replaced by have

l. 102 the word ‘part’ is missing

l.129 replace ‘faineancy’ by a more comprehensible word

l. 228 correct ‘hardmard’ by ‘Hadamard’

Reviewer 3 Report

The paper presents yet another approach to knowledge graph completion leveraging a pre-trained language model. The paper requires some minor polishing and would benefit from professional proofreading, but apart from that reads well is seems to be quite comprehensive.

Unfortunately, the authors choose a well-researched task, with a very competitive state-of-the-art (SOTA). In particular, [1] reports on the current SOTA on the FB15k-237 dataset, namely RESCAL-N3-RP, which outperforms the proposed approach on Hits@10, Hits@3, and Hits@1. Similarly, [2] reports on the current SOTA on WN18RR, where LP-BERT again outperforms the proposed approach on Hits@10, Hits@3, and Hits@1. These two results were the cornerstones of the paper, and without them, the paper lacks the necessary support to claim scientific advancement. In this light, I would recommend "reject with resubmission" - the paper as it is should not be published as the experiments indicate that there is no improvement over the current SOTA. However, should the authors be able to substantially improve the method and beat the SOTA, a revised version of the paper could be considered again for publication.

Minor remarks:

* The task is called "link prediction", not "linking prediction"

* "donates" -> "denotes" (in several places)

* Equations 3 and 4 are for We and Wp, yet the next paragraph talks also about We' and Wp'. What are these?

[1] https://paperswithcode.com/sota/link-prediction-on-fb15k-237
[2] https://paperswithcode.com/sota/link-prediction-on-wn18rr

Round 2

Reviewer 3 Report

The authors addressed all my remarks, including the major one about not beating the state of the art. In this light, I believe the paper is suitable for publication.